# A Lossless Compression Method for Chat Messages Based on Huffman Coding and Dynamic Programming

**Julián Moreno Cadavid *** and **Hernán Darío Vanegas Madrigal**

Departamento de Ciencias de la Computación y Decisión (Medellín), Universidad Nacional de Colombia—Sede Medellín, Medellín 050036, Colombia; hdvanegasm@unal.edu.co
*   Correspondence: jmoreno1@unal.edu.co

**Abstract:** There is always an increasing demand for data storage and transfer; therefore, data compression will always be a fundamental need. In this article, we propose a lossless data compression method focused on a particular kind of data, namely, chat messages, which are typically non-formal, short-length strings. This method can be considered a hybrid because it combines two different algorithmic approaches: greedy algorithms, specifically Huffman coding, on the one hand and dynamic programming on the other (HCDP = Huffman Coding + Dynamic Programming). The experimental results demonstrated that our method provided lower compression ratios when compared with six reference algorithms, with reductions between 23.7% and 39.7%, whilst the average remained below the average value reported in several related works found in the literature. Such performance carries a sacrifice in speed, however, which does not presume major practical implications in the context of short-length strings.

**Keywords:** data compression; dynamic programming; huffman coding; chat compression





## 1. Introduction

Handling the increasing amount of digital data generated every day is not an easy task. To address the aforementioned issue, there are at least two unmistakable solutions, using better hardware, or by using better software. A third solution would be a combination of the two. Although hardware manufacturers are always providing more robust solutions, it seems almost impossible to escape from the well-known consequences of Moore's law, that is, the amount of data grows at a higher rate than hardware can, unless some heretofore unknown technology is developed.

From the software perspective, compression is a solution, however, not simply any compression but specifically lossless compression. From a broad perspective, compression means encoding information using fewer bits than the original representation in order to reduce the consumption of critical resources, such as storage and bandwidth, or even for security reasons [1–10]. Compression can be either lossy or lossless. Lossy compression reduces bits by removing unnecessary or less important information, whereas lossless compression does so by identifying and eliminating statistical redundancy. As its name implies, no information is lost in lossless compression. In other words, lossless compression allows the original data to be entirely reconstructed from the compressed data, whereas lossy compression permits reconstruction of only an approximation of the original data.

Considering the above, in this paper, we present a lossless compression method based on a traditional algorithm: Huffman coding. However, we took a different path; instead of using variable length encoding for individual characters, we decided to use a dynamic programming approach for encoding substrings in an optimal way. In addition to this feature, our proposal involves two other important considerations. First, it focuses on the compression ratio, not on the compression speed. Second, it is not a general-purpose method. Instead, this method is intended for a particular kind of data: chat messages. In the very first line of this introduction, we talked about data volumes, and the chat

function is no exception to such volumes. Taking into consideration only two platforms, Goode [11] stated that, in 2016, Facebook Messenger and WhatsApp combined processed nearly 60 billion messages a day, three times more than the global volume of SMS messages.

We describe our method in two sections: first, we explain the preparation of the dictionary that will be used in the compression process. In this section, we use sample texts to obtain the most frequent characters and, later, the most frequent substrings. Then, we apply a filtering process to reduce the size of the dictionary, and we find an unambiguous way to encode each entry in the dictionary. Second, we explain the encoding process of a given text. Here is where the dynamic programming enters the scene. Finally, we compare the method against several alternatives.

## 2. Chat as a Form of Communication

Though ubiquitous, chatting is a relatively new form of communication that exhibits oral as well as written features. Holgado and Recio [12] defined a chat as an oral conversation over a written medium whose primary aim is real-time interaction with other people.

Now, what makes chat so uniquely fitted to our research purposes? That is, why did we focus only on chat for our compression method? There are at least two main features of chat compared with other written forms that we were principally interested in; the first is linguistic and the second is technical. From a linguistic perspective, chat contains a significant amount of non-normative uses [13–18], which makes using a standard, formal, dictionary approach not possible for compression purposes. For example, according to Llisterri [19], elision and epenthesis are two common phonetic deviations. Elision means the loss of sounds within words. It could be at the beginning of the word (apheresis), in the middle of the word (syncope), or at the end of the word (apocope). Some examples include "cause" for "because", or "fav" for "favorite". The opposite of elision is epenthesis, which is the addition of sounds at the beginning of the word (prosthesis), in the middle of the word (epenthesis), or at the end of the word (paragoge). Some examples include "Oo. . . oh" for "oh", "Wo. . . ow" for "wow", and "Helloo. . . " for "Hello".

There are also many other non-normative uses in chat. One of them is the graphical representation of an oral sound (onomatopoeia). Some examples include "hahaha" for laughter, or "weeeee" for happiness. Another deviation is the excessive use of abbreviations and acronyms. An abbreviation often used in the chat function is "CU" for "see you", whereas some examples of acronyms are "LOL" for "laugh out loud", or "BTW" for "by the way". There are also combinations of both, like "CUL8R" for "see you later". Perhaps the most representative lexical deviation is the use of emoticons. An emoticon, a term created from the combination of the words "emotion" and "icon", is a pictorial representation of a facial expression using punctuation marks, numbers, and letters, usually written to express a person's mood.

From the technical perspective, there is something very particular about chat: it is generally composed of small conversations. According to Xiao et al. [20], the average text length of chat messages stands between the tens and the hundreds of bytes. What this means, in the context of data manipulation, is that these conversations may be seen as a set of short length strings. Therefore, some compression methods that are based on long texts are not suitable.

## 3. Lossless Compression Algorithms and Huffman Coding

There are different types, or families, of lossless compression algorithms. One of these families is the dictionary-based methods, which build a dictionary of full words or substrings and assigns them a particular index or pointer code under the assumption that the same elements will appear as frequently in the future as they did in the past. Some examples are LZ77, LZ78, and LZW. Another family is the statistical methods, which analyze the most frequent characters and assign them shorter binary codes to obtain a shorter average coding length of the text. Some examples of these methods are Huffman Coding, Shannon–Fano coding, and arithmetic coding.

Huffman coding was defined in 1952 by David Huffman [21] and has been used as the basis for several compression methods for multiple data types. In this algorithm, the characters of a particular alphabet are converted to binary codes, where the most common characters coming from some input have the shortest binary codes, whereas the least common have the longest. In order to do so, and, as the "heart" of the algorithm, it builds a binary tree that allows for coding each character without ambiguity. This way, the coding process may be seen as the sum of two phases—the construction of the frequency table and the construction of the binary tree.

The construction of a frequency table is a straightforward process, which consists of counting the occurrences of each individual character in a sample text. As a result, a table with pairs [character, frequency] is obtained. Then, the construction of the binary tree works as follows:

Step 1: Move each pair of the previously obtained table into a list of nodes. Each node has four attributes: symbol, frequency, left, and right. The first two correspond to the character and frequency values of the table. The last two aid in the construction of the binary tree structure and are initially set to NULL in this step for all nodes.

Step 2: Select the two nodes *A* and *B* with the lowest frequency from the list.

Step 3: Create a parent, empty, node *C* to "merge" them, setting its symbol to the concatenation of *A* and *B* symbols, its frequency to the sum of *A* and *B* frequencies, its left child as *A*, and its right child as *B*.

Step 4: Remove A and B from the list and add C.

Step 5: Repeat steps 2 to 4 until a single node remains in the list.

This iterative procedure guarantees a single and optimized binary code for each character by simply traversing the binary tree from the root to the leaf nodes: each move to the left child adds a '0' and each move to the right child adds a '1' to corresponding node coding. While doing so, a table with pairs [character, binary code] is created. This table is called the codes dictionary. As consequence of this process, characters with lower frequencies are "sacrificed" in the earlier iterations and, therefore, end up with larger binary codes, whereas characters with higher frequencies are "saved" for the later ones and end up with shorter codes.

For decoding, i.e., the opposite process of turning a binary array into the original data, the next procedure must be followed:

Step 1: Start at the beginning of the binary code.

Step 2: Read the next bit and form with it a subarray of bits.

Step 3: Look into the codes dictionary to find if such an array has a character translation. If so, append that character to the decoded data and clear the subarray. If not, read the next bit and append it to the subarray.

Step 4: Repeat steps 2 and 3 until reaching the end of the binary code.

## 4. Proposed Method

In the last 15 years, there have been several studies on methods to compress text [22–30]. Some of them were based on the Huffman coding algorithm [27,28], whereas others were intended particularly for chat messages [26] or for other types of short-length messages [22,25,29]. Some focused on compression speed [24,29], whereas others, as in our proposal, focused on effectiveness [23]. What makes our proposal special? Essentially, we introduce two significant changes concerning the standard Huffman algorithm. First, our method does not use only individual characters of the given alphabet to construct the codes dictionary. Instead, it considers multiple length strings. By doing so, single characters become a particular case of the algorithm, i.e., strings of length one. Our claim here is that chats, in general, are composed of common substrings, whose frequencies are large enough that it is worth coding them as a whole and not as individual characters. This way, the coding process is no longer trivial. The second change is that an optimization process

is used to determine the best way to divide a given text into substrings, to ensure that the corresponding coding has the minimum length.

In this sense, the method proposed may be considered hybrid, algorithmically speaking because it combines a greedy algorithm with a dynamic programming one. The greedy algorithm (Huffman coding) allows for the construction of the codes dictionary, whereas the dynamic programming algorithm allows for determining the optimal way to codify a text accordingly.

Considering these two issues, Figure 1 summarizes the method proposed where three main processes are clearly identified: pre-processing, coding, and decoding.

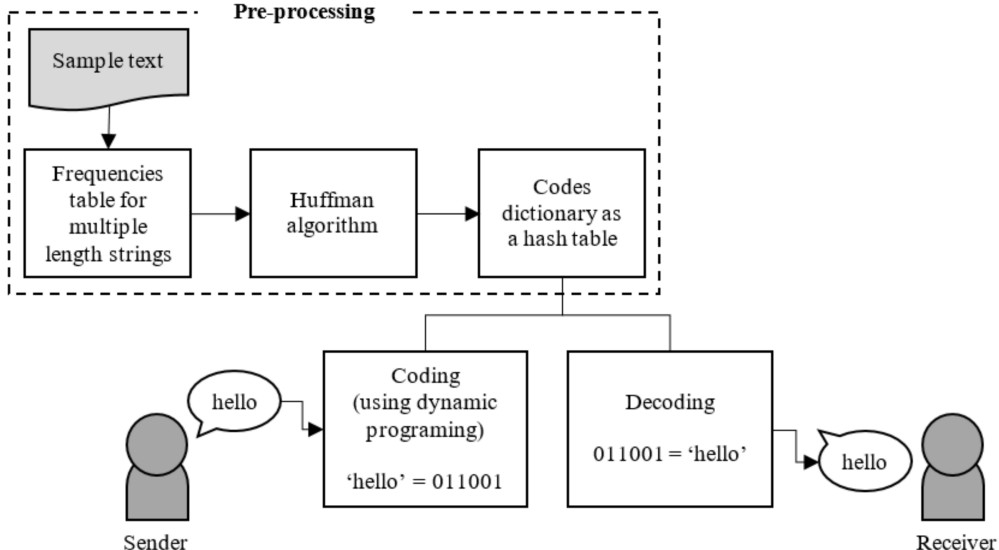

**Figure 1.** Method summary.

### 4.1. Pre-Processing

During pre-processing, three sub-tasks are performed. However, before this, a sample text is necessary, which must be large and meaningful enough. On the one hand, being meaningful means that it must represent the context in which the text compression will take place. One obvious condition is that the language must be the same. However, it must also be as coherent as possible. If chat messages are the main goal, actual chat records should preferably be used. On the other hand, being sufficiently large means that it should contain hundreds, or hopefully even thousands or dozens of thousands of lines. This guarantees that the frequency count is statistically significant.

Considering the sample text, the first sub-task begins with counting the frequency of each single character of the considered alphabet. Until this point, this task works exactly the same as the one described in Section 3 for the Huffman algorithm. It is necessary, however, to define precisely what that alphabet will be. In this proposal, we used a subset of the UTF-8 standard and, more specifically, all the letters a–z, A–Z, plus the digits 0–9.

We also considered a special character for a new line; some characters needed for writing Spanish, i.e., the letters ñ, Ñ, and accented vowels, as well as several characters for emoticons and other purposes in chats. In total, as shown in Table 1, 112 characters were considered; therefore, the frequency count for individual characters had 112 entries. Those characters that did not appear even once in the sample text appear in the table with a frequency of zero. An important consideration for the algorithm is that the previous characters' subset can be changed depending on the considered language and context.

**Table 1.** Characters considered in the alphabet.

| (space) | ! | " | # | $ | % | & |
|---|---|---|---|---|---|---|
| ' | ( | ) | * | + | , | - |
| . | / | 0 | 1 | 2 | 3 | 4 |
| 5 | 6 | 7 | 8 | 9 | : | ; |
| < | = | > | ? | @ | A | B |
| C | D | E | F | G | H | I |
| J | K | L | M | N | O | P |
| Q | R | S | T | U | V | W |
| X | Y | Z | [ | \ | ] | ^ |
| _ | a | b | c | d | e | F |
| g | h | i | j | k | l | m |
| n | o | p | q | r | s | t |
| u | v | w | x | y | z | { |
| \| | } | ~ | ¡ | ¬ | º | ¿ |
| Á | É | Í | Ñ | Ó | Ú | Ü |
| á | é | í | ñ | ó | ú | ü |

From this point, the standard procedure was no longer followed. After establishing these 112 values, we counted the frequencies of all substrings of length two. Considering an arbitrary sample text, this would give us up to $112^2$ new entries. Then, we did the same for substrings of length three, obtaining up to $112^3$ new entries, and so on. Considering the exponential effect of doing so, we selected a value of six for the maximum substring length. Even though such a value may seem short, it makes perfect sense considering that there would be relatively few longer substrings, at least in Spanish and English, with a statistically significant frequency. Even with that simplification, the total number of entries could be as large as $1.99 \times 10^{12}$.

We then reduced this table considering only the first 112 one-character entries, plus the remaining $k'$ more frequent entries. For the sake of clarity, we named $k = k' + 112$ as the size of the dictionary D. The first 112 entries are needed, even if some of them have a frequency of zero because, without them, it would not be possible to code any arbitrary text within the controlled alphabet. As will be evidenced later during the experimental results, choosing a suitable value of $k$ is critical because it alters the ratio of compression. The larger the $k$, the more possibilities for coding the same string, however, the more complexity to obtain the optimum.

Once we established this table, we proceeded with the second sub-task, which was running the Huffman algorithm. This procedure does not differ from the one described in Section 3. However, instead of a binary tree for characters, one for substrings is obtained.

Once we have this binary tree, the third sub-task refers to "translating" it into a table of pairs of substring–binary code. Again, this may be done by traversing the binary tree from the root to the leaf nodes: each move to a left child adds a '0', and each move to a right child adds a '1' to corresponding node coding. Here, we introduce another feature: this table is stored in a data structure that guarantees a fast search of a value given a key during the coding process. In this case, the keys refer to the substrings, whereas the values refer to the corresponding binary codes. Alternatives of efficient data structures for that purpose include self-balanced binary search trees, crit-bit trees, and hash tables, among others [31]. From those, we chose a hash table for our implementation.

Algorithmically speaking, the full pre-processing has a time complexity of $O(S)$ for the frequency count, with S as the length of the sample text, plus $O(k \cdot \log(k))$ for obtaining the coding as a binary tree using the Huffman algorithm. Even if the values of S and k are large, the full process must be performed only once and, in all cases, prior to the actual compression; therefore, it should not be a concern.

*4.2. Coding*

Once the binary tree is obtained in the standard Huffman algorithm, the coding process is practically trivial. In our case, however, such a process is the core of the proposal and is not trivial at all. If, for example, with the standard algorithm, the text "yes" needed to be coded, what we would have is the binary codes for the three characters 'y', 'e', and 's'. However, considering multiple length strings instead of just individual characters, we might have also the substrings "ye", "es", or "yes". In this case, there would be, in total, four different ways to divide such a text. In general, given a string composed of $m$ characters, there would be up to $2^{m-1}$ different ways to split it depending on all the sub-stings that are part of the dictionary. For instance, with a word like "dictionary", we could have up to 512 options.

Considering the aforementioned issue, the coding process transforms into an optimization problem. More specifically, given a text $T$ of length $n$ we need to find, from up to $2^{n-1}$ possibilities, the one that minimizes the total length of the obtained binary code. In other words, we must find the partitions $T_1, T_2, \ldots, T_j$, such as $T = T_1 + T_2 + \cdots + T_j$, and the sum of the code lengths of such partitions is minimal. This problem may be solved using a dynamic programming approach as presented next.

Step 1: Find the optimal way to code all consecutive substrings of length 1 from $T$. Such solutions are trivial, we simply have to search the corresponding individual characters in the dictionary $D$ obtained during pre-processing.

Step 2: Find the optimal way to code all consecutive substrings of length 2 from $T$. For this, we need to compare if such a two character substrings is better than adding the optimal partitions found in the previous step. Here, "better" indicates that (i) such a substring is in $D$, and (ii) the corresponding code is shorter.

Step 3: Find the optimal way to code all consecutive substrings of length 3 from $T$. For this, we need to compare if such a three character substrings is better than adding the optimal partitions found in the previous steps.

Step $h$ ($4 \leq h \leq n$): Find the optimal way to code all consecutive substrings of length $h$ from $T$. For this, we need to compare if such an $h$ character substring is better than adding the optimal partitions found in the previous steps.

A more detailed description of this procedure is presented in Algorithm 1, which is based on the solution for the matrix chain multiplication problem described by Cormen et al. [32].

As in the matrix chain multiplication algorithm [32], this yields a running time of $O(n^3)$, due to the triple loop in lines 8 to 27, and requires $O(n^2)$ space to store the matrices. It is, however, insufficient to solve the coding process. Once we obtain matrices $A$ and $B$, we must find the indexes in which we should divide $T$ (until now, we obtained the optimal length but not the corresponding partitions). This procedure is performed by a backtracking algorithm (Algorithm 2) that takes the matrix $B$ as a parameter and finds the required indexes.

The result of this algorithm is a list $L$ with the indexes of the optimal partition of $T$. With those indexes, the corresponding substrings may be searched in $D$ and, therefore, the whole binary coding of $T$ may be found. This last algorithm yields a running time of $O(n^2)$, and thus it does not add extra complexity to the dynamic programming approach.

The whole optimization process through the dynamic programming approach guarantees, at least, the same compression performance as the standard Huffman algorithm. This worst-case scenario would exist only in the case that no substrings at all appear in the text with a significant frequency, i.e., almost as "white noise", something not common in written text, and definitely not common in chat messages.

---

**Algorithm 1** Optimal partition

---

**function** OPTIMALPARTITION(Text $T$, Dictionary $D$)

    Let $A$ be a matrix with order $n \times n$

    Let $B$ be a matrix with order $n \times n$

    **for** $i = 0$ to $n - 1$ **do**

        $A[i][i] = D.getValue(T.charAt(i)).length()$

        $B[i][i] = i$

    **end for**

    **for** $l = 2$ to $n$ **do**

        **for** $i = 0$ to $n - 1$ **do**

            Let $j$ be an integer

            $j = i + l - 1$

            Let $s$ be a String $s = T.substring(1, j + 1)$

            **if** $D.containsKey(s)$ **then**

                $A[i][j] = D.getValue(s).length$

            **else**

                $A[i][j] = \infty$

            **end if**

            **for** $k = j - 1; k \geq i; k - -$ **do**

                Let $q$ be an integer

                $q = A[i][k] + A[k + 1][j]$

                **if** $q < A[i][j]$ **then**

                    $A[i][j] = q$

                    $B[i][j] = k$

                **end if**

            **end for**

        **end for**

    **end for**

**end function**

---

---

**Algorithm 2** Backtrack

---

   **function** BACKTRACK(Matrix $B$, List $L$, $i$, $j$)

      **if** $i == j - 1$ **then**

         **if** $B[i][j] \neq j$ **then**

            $L.add(B[i][j])$

         **end if**

      **else**

         **if** $B[i][j] \neq j$ **then**

            $L.add(B[i][j])$

            BACKTRACK$(B, L, i, B[i][j])$

            **if** $c[i][j] + 1 < B.length$ **then**

               BACKTRACK$(B, L, B[i][j] + 1, j)$

            **end if**

         **end if**

      **end if**

   **end function**

---

*4.3. Decoding*

Considering that, independently of how the coding occurred, the binary codes in D are unambiguous; therefore, the decoding process may be performed in the exact same way as in the standard Huffman algorithm, i.e., sequentially through the coded string searching for the corresponding translations in the dictionary D. However, and as a consequence of the introduced changes, differently to the standard algorithm, the coded text is not decoded necessarily character by character but substring by substring.

**5. Experimental Results**

To demonstrate the usefulness of the method proposed, named HCDP for Huffman Coding + Dynamic Programming, we performed a comparison with several alternatives. However, before doing so, it was important to determine the comparison criterion. Here, we chose the Compression Ratio, a very common measure in the compression context [2–7]:

$$CR = \frac{\text{Bytes of the compressed file}}{\text{Bytes of the original file}}. \tag{1}$$

From the start, we discarded other conventional measures, like running time because we are well aware that an $O(n^3)$ complexity is far from being promising for that matter.

For the comparison, we selected six algorithms that are often used for compression benchmarking: LZMA, LZW, PPM, PAQ8, gzip, and bzip2. In addition to these, we attempted to use algorithms specifically designed for the compression of chat or, at least, of short text messages. From the four mentioned in Section 4, only one [29] had the code available: b46pack (https://github.com/unixba/b64pack, accessed on 28 February 2021), which has a fixed compression ratio of 0.75 for messages of up to 213 characters. Of the other three, one [26] used LZW, whereas the other two [22,25] used their own algorithms for which, although not available, the authors reported average compression ratios of 0.4375 and 0.445, respectively.

As sample texts, we used several well-known literary works: Don Quixote by Miguel de Cervantes Saavedra, Alice in Wonderland by Lewis Carroll, Paradise Lost by John Milton, The House of the Spirits by Isabel Allende, and A Hundred Years of Solitude by Gabriel García Márquez. Some of these texts were found in the Canterbury corpus (http://corpus.canterbury.ac.nz, accessed on 28 February 2021), a collection of files created in 1997 at the University of Canterbury and intended for use as a benchmark for testing lossless data compression algorithms.

The method proposed was intended for chat compression; therefore, we expected sample texts from chat records. This is difficult, however, mainly because chat messages are usually private. An alternative could have been saving our own private chat messages, but that, in the best case, would have introduced biases in the text. To solve this issue, we found a method for obtaining chat messages from multiple users in a Learning Management System (LMS) of our University, with the explicit permission of those users.

Using such sample texts, we tested the compression ratios of each method. For the comparison, we prepared a dictionary D for each text, and then we compressed them one by one, breaking them down into blocks of up to 100 words. The compression ratio was calculated for each text block, and then they were averaged. This ratio in our proposal does not include the size of the dictionary D because this remains saved in the compressor/decompressor, not in the compressed files. The results are shown in Table 2.

**Table 2.** Compression ratios.

| Sample Text | Size in Bytes | Compression Ratio | | | | | | |
|---|---|---|---|---|---|---|---|---|
| | | **LZMA** | **LZW** | **PPM** | **PAQ8** | **gzip** | **bzip2** | **HCDP** |
| 1: A Hundred Years of Solitude * | 827,014 | 0.6734 | 0.6084 | 0.5683 | 0.5799 | 0.5991 | 0.5991 | 0.4131 |
| 2: Alice in Wonderland | 152,086 | 0.6920 | 0.6181 | 0.6065 | 0.6110 | 0.6218 | 0.6640 | 0.4172 |
| 3: Don Quixote * | 323,156 | 0.6730 | 0.6065 | 0.5607 | 0.5820 | 0.5998 | 0.6219 | 0.4068 |
| 4: House of the Spirits * | 981,187 | 0.6804 | 0.6104 | 0.5745 | 0.5868 | 0.6048 | 0.6225 | 0.4250 |
| 5: LMS chat * | 316,292 | 0.6379 | 0.5943 | 0.5685 | 0.5709 | 0.5865 | 0.6431 | 0.4339 |
| 6: Paradise Lost | 481,859 | 0.7185 | 0.6269 | 0.6345 | 0.6218 | 0.6380 | 0.6671 | 0.4456 |
| Mean | | 0.6792 | 0.6108 | 0.5855 | 0.5921 | 0.6083 | 0.6393 | 0.4236 |

* Text in Spanish.

In all cases, the HCDP method performed better, demonstrating the lowest compression ratio. More specifically, HCDP had, on average, a 37.6% lower compression ratio when compared with LZMA, 30.6% lower compared with LZW, 27.7% lower compared with PPM, 28.5% lower compared with PAQ8, 30.4% compared with gzip, and 33.4% lower compared with bzip2.

Considering only our context of interest, i.e., chat compression, HCDP also had the best compression ratio in the LMS chat sample text. It is curious, though that this text had the second largest ratio for HCDP, whereas, for all the others, its ratio was the lowest or the second lowest.

With regard to other algorithms designed specifically for the compression of short text messages, including chat, HCDP had, on average, a 43.5% lower compression ratio when compared with b64pack and 30.6% lower compared with [26]. It was also lower by 3.2% and 4.8% compared with the values reported in [22,25]. In these two last cases, we considered the values reported by the authors using their own sample texts.

In keeping with the findings, Table 3 presents the most frequent characters in each sample text. Here, there are no surprises. The most frequent character by far, despite the language, was whitespace. This implies that this character has the shortest binary code in both methods—standard Huffman and HCDP. The vowels A, E, O, and I, jointly with the consonants T, H, N, S, and R were the next most frequent characters in the two English sample texts. In the Spanish sample texts, there was less homogeneity, although, in all cases, the vowels A, E, and O were the most frequent characters.

**Table 3.** Most frequent characters in the sample texts.

| Rank | text1 | text2 | text3 | text4 | text5 | text6 |
|------|-------|-------|-------|-------|-------|-------|
| 1 | " " 137,842 | " " 28,900 | " " 54,077 | " " 166,178 | " " 34,677 | " " 81,727 |
| 2 | "a" 84,238 | "e" 13,381 | "e" 32,568 | "a" 102,875 | "a" 31,306 | "e" 45,114 |
| 3 | "e" 79,253 | "t" 10,212 | "a" 29,296 | "e" 91,143 | "e" 24,668 | "t" 29,794 |
| 4 | "o" 55,751 | "a" 8149 | "o" 22,760 | "o" 65,236 | "o" 23,068 | "o" 26,309 |
| 5 | "n" 45,361 | "o" 7965 | "s" 17,940 | "s" 56,228 | "s" 14,399 | "a" 24,823 |
| 6 | "s" 45,068 | "h" 7088 | "n" 15,797 | "r" 51,428 | "l" 13,462 | "n" 24,692 |
| 7 | "r" 44,534 | "n" 6893 | "r" 14,756 | "n" 50,526 | "n" 11,230 | "h" 23,690 |
| 8 | "l" 38,413 | "i" 6778 | "l" 13,570 | "l" 44,302 | "i" 11,050 | "s" 23,018 |
| 9 | "i" 36,218 | "s" 6277 | "d" 12,214 | "i" 38,684 | "r" 9194 | "r" 22,813 |
| 10 | "d" 33,748 | "r" 5293 | "u" 11,754 | "d" 37,098 | "u" 7992 | "i" 22,350 |

When considering substrings rather than individual characters, especially in the LMS chat as shown in Table 4, several interesting issues appeared. First, there were several cases in which the method was able to take advantage of frequent substrings to improve the compression ratio. For example, coding the substring "jajaja" as a whole in that sample text, resulted in being more efficient (10 bits) than coding "ja" three times (27 bits), or coding each character individually (42 bits). This similarly occurred with the substring "aaaa" (8 against 24 bits), with the substring ":v" (13 against 20 bits), and with the substring ":(" (14 against 24 bits).

**Table 4.** Examples of LMS chat substrings.

| String | Frequency | Code Length |
|--------|-----------|-------------|
| a | 34,677 | 6 |
| j | 4776 | 8 |
| aaaa | 3028 | 8 |
| ja | 2612 | 9 |
| v | 2583 | 9 |
| hola | 1933 | 9 |
| jajaja | 770 | 10 |
| : | 467 | 11 |
| ( | 105 | 13 |
| :v | 96 | 13 |
| :( | 63 | 14 |

These substrings also highlight the importance of considering chat differently than formal writing forms: "jajaja" as a common form of onomatopoeia (laugh), "aaaa" as a common form of epenthesis, and ":v", ":(" as common emoticons.

To explain these results with a particular example, and to further clarify the coding process of the proposed algorithm, let us consider the next message: "hola, como estas? juguemos mas tarde", which is the Spanish version, with the usual spelling mistakes, of "hi, how are you? Play later". If we use, as a reference text, the LMS chat records with the corresponding codes dictionary, the optimization process for the string partition would result in:

"hola"—100111000
", "—1011100100011
"como"—11110000101
"estas"—010110101101
"?"—100100011
" jug"—11111101000110
"ue"—00010000

"mos"—1010101011
" mas "—0010100000000
"tar"—0111100100110
"de"—111000010

In this way, the entire binary code array contains 121 bits, compared with the 288 bits (36 characters ∗ 8 bits per character) of the standard UTF8 coding. This implies a compression ratio of 121/296 = 0.4201.

Finally, and as we stated earlier in Section 4, one important parameter in the proposed method is the size of the dictionary. The results presented in Table 2 were obtained with a value of such a parameter after a calibration process. We ran the method for all sample texts starting with a size of 256. This initial value stands for the size of the dictionary considering the 112 individual characters plus 144 substrings. This sum, 256, was chosen for representing exactly one byte in non-extended ASCII. From that, we ran the method over and over enlarging that size in factors of 256 as shown in Figure 2.

When the dictionary size increased, the compression ratio decreased. However, that decrease became slower each time. Therefore, we defined stopping criterion for when the variation of the mean compression ratio dropped below 0.001, which implies an improvement in this value of less than 0.1%. This way, the resulting size was 7424 elements. Although that value implies that we require a large dictionary, it does not affect the compression ratio because it does not have to be saved along with the compressed files, but remains in the compressor/decompressor.

Considering that the mean of the compression ratio suggested a logarithmic shape, we performed a regression, finding the next equation:

$$\bar{r} = -0.0321 \ln(k) + 0.5198, \tag{2}$$

with a determination coefficient $R^2 > 0.99$, this equation serves as a good predictor of the method performance. However, some kind of equilibrium must be achieved. The larger the dictionary, the more opportunities to treat larger substrings as unique codes, but also the larger average code length. This is why a stopping criterion, such as the one defined, was relevant.

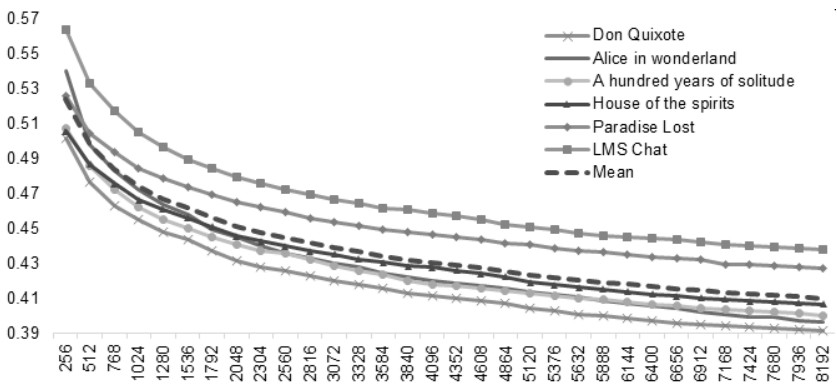

**Figure 2.** Code dictionary size vs. the compression ratio.

## 6. Conclusions

Lossless compression is a crucial matter in computing, particularly when resources, like bandwidth or storage, are expensive. What makes the proposal presented in this paper of interest is that we focused on a specific context—chat messages—and exploited the inherent features to obtain better performance. In particular, we considered that (1) chat consists of short text messages, and (2) it is full of non-normative uses of written language that repeat frequently.

Similar to other approaches [27,28], our approach was based on the Huffman coding algorithm; however, we introduced two major changes. First, we considered multiple

length substrings rather than individual characters, and, second, we included an optimization procedure based on dynamic programming. This is why we named our method HCDP (Huffman Coding + Dynamic Programming). These two modifications together made a significant difference in terms of the compression ratios but with a sacrifice in speed.

Whereas the traditional Huffman algorithm and most of its variants yield a $O(n^2)$ running time, HCDP yields $O(n^3)$. However, in computational terms, this represents a considerable gap, ultimately there is no practical repercussion due to the short nature of text message chats and to the fact that the entire compression/decompression process is carried out on the client side.

Our experimental results demonstrated that HCDP performed better than the six general purpose compression algorithms: LZMA, LZW, PPM, PAQ8, gzip, and bzip2. Considering six different sample texts, the compression ratio of HCDP, with an average of 0.4236, was lower in all cases with reductions between 23.7% and 45.8% (see Table 2). Due to the unavailability of the corresponding source codes, these results cannot be directly compared with related works intended specifically for chats or other types of short text messages, except for [26], which used LZW, and [29], which exhibited a fixed compression ratio of 0.75. Instead, in others, such as [22,25], they can be compared with the reported average values, which are 0.4375 and 0.445, respectively. In these last two cases, HCDP represents a reduction of only 3.2% and 4.8%, respectively, but a reduction nonetheless.

A good part of the success of our proposal is that, in written narratives, and particularly in chats, in addition to the frequent use of characters of the language, there is a recurrent use of substrings, words, and even expressions that, once codified in binary as a whole, allow the optimization procedure to save even more bits (see Table 4).

Another particular feature that is in favor of the proposed method is that the pre-processing required for obtaining the code dictionary may be done according to the specific context and may, therefore, optimize the compression ratio. For example, if someone is developing a chat messaging service for a dating platform, they could attempt to collect chat records in that context so that the corresponding dictionary would be optimized for that matter.

The same would occur in an online game, in a sports app, and so on. In fact, as a future work, we intend to repeat the performed experiment with much larger, and even more specific, chat datasets. A perfect case would be, for example, chats from a Massive Multiplayer Online Game (MMOG). Although access to such records in the case of popular commercial titles is restricted, we could use academic alternatives, but not before obtaining the respective informed consent and guaranteeing the corresponding protection of privacy for the participants.

Finally, and to facilitate further comparisons, the corresponding implementation in Java is available for anyone to use in https://github.com/hernan232/HCDP (accessed on 28 February 2021).

**Author Contributions:** Data curation, H.D.V.M.; investigation, J.M.C.; software, H.D.V.M.; writing—original draft, H.D.V.M.; writing—review & editing, J.M.C. Both authors have read and agreed to the published version of the manuscript.

**Funding:** This research received no external funding.

**Institutional Review Board Statement:** Not applicable.

**Informed Consent Statement:** Not applicable.

**Data Availability Statement:** Authors can confirm that all relevant data are included in the article.

**Conflicts of Interest:** The authors declare no conflict of interest.

## Abbreviations

The following abbreviations are used in this manuscript:

| | |
|---|---|
| ASCII | American Standard Code for Information Interchange |
| CR | Compression Ratio |
| HCDP | Huffman Coding + Dynamic Programming |
| LMS | Learning Management System |
| LZ | Lempel–Ziv |
| LZMA | Lempel–Ziv–Markov chain Algorithm |
| LZW | Lempel–Ziv–Welch |
| MMOG | Massive Multiplayer Online Game |
| PPM | Prediction by Partial Matching |
| SMS | Short Message Service |
| UTF | Unicode Transformation Format |

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
