# Peer review of "A Lossless Compression Method for Chat Messages Based on Huffman Coding and Dynamic Programming"

_computers, doi:10.3390/computers10030028_

Round 1

Reviewer 1 Report

This paper proposes an approach focuses on a lossless data compression method focused on a particular kind of data, namely, chat messages. As authors mentioned those are mostly non-formal, short-length strings. Thus they provide a combined approach based on two different algorithms one greedy algorithm (Huffman coding) and a dynamic programming. A strength is the experiments conducted that show that proposed method provides lower compression ratios than other six reference7algorithms, with hopeful reductions. However, I missed a clear reasoning why those six algorithms were chosen.

Additionally, authors declare a sacrifice in speed in the context of short-length strings, which really delimits the optimal application scenario of this approach.

Regarding the related works I missed some relevant publications as:

“Breaking MPC implementations through compression” and “Entropy analysis to classify unknown packing algorithms for malware detection”.

Reviewer 2 Report

  1. There are numerous places in the text with English grammatical errors. The authors should be full checking for grammar and mistakes to meet the quality of Journal.

  1. Add the list of symbols.

  1. Most equations in the paper, the authors don't mention to any references. Add citation to the equations which took from references.

  1. Add a new flowchart to explain the details of the selected approach instead of Fig. of Method summary.

  1. 5. There are many research papers study the same problem that investigated in the present paper. What is exactly the new point of this work?

The authors should focus to clarify this issue in the paper.

  1. The meaning of the conclusions is unclear, and there are grammatical errors. The authors should think over the real significance of their results and try to rewrite this section to improve understanding of the conclusions.

Reviewer 3 Report

My major comment is that the experiment in Section 5 seems to be inappropriate. The algorithm in this manuscript is designed for compressing chat messages. Because the authors list some prior works of a compression method for chat messages (e.g., [18]--[26]) in lines 121--123 in the manuscript, they should compare their proposed method with these studies in the experiment, especially in the compression experiment for chat messages. Nevertheless, in the experiment in Section 5, the authors compare their method with general compression algorithms such as LZW and gzip.

I list some minor comments in what follows:

1. There are spelling inconsistencies in the manuscript. For example, ``Section 3'' in line 148, ``sectiion 3'' in line 172, and ``Part B, section IV'' in line 329.

2. There are typos in the manuscript. For example, ``we includes'' in line 374.

3. The source of Reference [1] is missing, i.e., Reference [1] only contains the author's name, title, and year.

Round 2

Reviewer 2 Report

The authors made all required corrections.

The paper now is accepted without any changes.